

# Assigning confidence scores to homoeologs using fuzzy logic

Natasha M. Glover[1,2,3], Adrian Altenhoff[1,4] and
Christophe Dessimoz[1,2,3,5,6]

[1] SIB Swiss Institute of Bioinformatics, Lausanne, Switzerland
[2] Center for Integrative Genomics, University of Lausanne, Lausanne, Switzerland
[3] Department of Computational Biology, University of Lausanne, Lausanne, Switzerland
[4] Department of Computer Science, ETH Zurich, Zurich, Switzerland
[5] Department of Genetics, Evolution, and Environment, University College London, London, UK
[6] Department of Computer Science, University College London, London, UK

## ABSTRACT

In polyploid genomes, homoeologs are a specific subtype of homologs, and can be thought of as orthologs between subgenomes. In Orthologous MAtrix, we infer homoeologs in three polyploid plant species: upland cotton (*Gossypium hirsutum*), rapeseed (*Brassica napus*), and bread wheat (*Triticum aestivum*). While we can typically recognize the features of a "good" homoeolog prediction (a consistent evolutionary distance, high synteny, and a one-to-one relationship), none of them is a hard-fast criterion. We devised a novel fuzzy logic-based method to assign confidence scores to each pair of predicted homoeologs. We inferred homoeolog pairs and used the new and improved method to assign confidence scores, which ranged from 0 to 100. Most confidence scores were between 70 and 100, but the distribution varied between genomes. The new confidence scores show an improvement over our previous method and were manually evaluated using a subset from various confidence ranges.

## INTRODUCTION

Polyploidy is an important and widespread phenomenon within the plant kingdom (*De Bodt, Maere & Van De Peer, 2005*), and redundancy at the level of chromosomes implies redundancy at the level of genes. More specifically, homoeologs are defined as genes which originated by a speciation event, diverged, and came back together some time later via allopolyploidization (*Glover, Redestig & Dessimoz, 2016*). (See Table 1 for definitions of terms related to polyploidy.) The homoeologous relationships between subgenomes of a polyploid can be used in order to determine the structural, genetic, and evolutionary results of polyploidization.

Since 2015 we have included pairwise homoeolog predictions between subgenomes in Orthologous MAtrix (OMA), which is a method and database for inferring evolutionary relationships (*Altenhoff et al., 2015*). With other methods, homoeologs are usually predicted using a best bidirectional hit approach, sometimes in combination with an added requirement of positional conservation, that is, synteny (*Dewey, 2011*).

Corresponding author
Natasha M. Glover,
natasha.glover@unil.ch

**Table 1 Definitions of relevant terms related to biology.**

| | |
|---|---|
| Polyploidy | Having more than two sets of homologous chromosomes; the result of genome doubling. |
| PAM units | Point accepted mutation. A measure of evolutionary distance; the amount of amino acid substitutions per 100 amino acids of a protein sequence. One PAM unit means that 1% of the amino acids were replaced since the divergence of the two protein sequences. |
| Homoeolog | Genes of an allopolyploid which started diverging by a speciation event, and were brought back to the same genome via a hybridization event. They can be thought of as orthologs between subgenomes. |
| Allopolyploid | A species which has more than one set of homologous chromosomes due to a whole genome duplication via hybridization. |
| Subgenome | One of the genome sets in a polyploid. |
| Synteny | The degree of gene position conservation between two diverging segments of chromosomes, in this case between two homoeologous chromosomes. |
| Evolutionary distance | The amount of divergence between two protein sequences. |
| Total copy number | An assessment of the amount of duplication for a given homoeolog pair. In this paper, it is the sum of the homoeologs for both genes of the pair. |

However, one-to-one correspondence or synteny among homoeologs may not hold for several reasons, such as lineage-specific duplications or small-scale translocations. In OMA, we do not rely on synteny for homoeolog inference, and we also allow for duplications after the hybridization event.

However, relaxing one-to-one and synteny criteria makes it harder to distinguish correct from incorrect calls. Furthermore, because of the redundancy and size, polyploid genomes can be difficult to assemble and annotate. For instance, *Triticum aestivum* (bread wheat) remains in a highly fragmented survey state, consisting of scaffolds rather than fully-assembled chromosomes (*International Wheat Genome Sequencing Consortium (IWGSC), 2014*; *Clavijo et al., 2017*). This previously motivated us to classify homoeolog predictions as "high" vs. "low" confidence based on chromosome matching (global synteny) (*Altenhoff et al., 2015*). For example, if one homoeolog of a pair was on chromosome 3B while the other homoeolog is on chromosome 3A, this pair was considered high confidence because they belong to the chromosome group 3. While this holds true most of the time due to the relatively short divergence time between subgenomes, this is coarse, and can be misleading in the presence of chromosomal rearrangements or small-scale translocations.

There have been several methods reported which yield quantitative confidence scores for ortholog or paralog predictions. For example, InParanoid assigns confidence scores to in-paralogs on a scale from 0 to 100 depending on how distant the predicted inparalog sequence is from the "main" ortholog. Additionally, InParanoid confidence scores are assigned to orthologous groups based on a technique that assigns a higher score to potential ortholog sequences that have much better bootstrap value than competing ortholog sequences (*Remm, Storm & Sonnhammer, 2001*). Several meta-methods for predicting orthology (i.e., those that combine the results of many different orthology inference algorithms) report confidence scores. For example, MetaPhOrs gives confidences scores based on the number of independent sources the orthology prediction was found in, as well as scores based on the consistency for which scores are computed using their method (*Pryszcz, Huerta-Cepas & Gabaldón, 2011*). The Drosophila RNAi

**Table 2 Definition of terms related to fuzzy logic.**

| | |
|---|---|
| Fuzzy logic | A type of mathematical logic based on natural language where truth is considered on a continuous scale as degrees of truth rather than binary true or false. Fuzzy logic resembles human reasoning and intuition because it uses classes with unsharp boundaries, defined with natural language. |
| Control system | The mathematical models which make up the fuzzy inference process. |
| Universe of discourse | A set of all possible values defined for a fuzzy input or output. |
| Membership function | A function, normally visualized graphically, which denotes a fuzzy set. The membership function represents the degree (between 0 and 1) to which an element in the universe of discourse belongs. The membership functions represent fuzzy sets also represent linguistic variables, which overlap so that an input may belong to two categories, each to a certain degree. |
| Fuzzification | The process of translating a crisp input to a fuzzy one. This is the first step of the fuzzy inference process, where the crisp input gets mapped to its fuzzy set based on the membership functions. |
| Defuzzification | The process of converting the fuzzy output derived from the fuzzy inference process to a crisp output. |
| Fuzzy rules | A set of "if … then" rules needed for mapping the fuzzy input to the fuzzy output. These rules are based on a human's expertise, knowledge, and intuition. The fuzzy rules are defined and stored in the lookup table. |
| Crisp input or output | Input or output which has a quantitative value, limited to the range of the universe of discourse. |
| Fuzzy inference process | The fuzzy inference process consists of taking the crisp input, fuzzifying it, combining it with the fuzzy rules, and defuzzyfying, resulting in a crisp output. |
| Fuzzy set | A set with unsharp boundaries, as defined by the membership function. Fuzzy sets allow for its members to belong to more than one set the same time, to some partial degree. |

Screening Center Integrative Ortholog Prediction Tool gives confidence scores based on not only the number of independent algorithms predicting each pair, but also weighted to reflect the functional similarity, as shown by GO semantic similarity (*Hu et al., 2011*). A recent method called WORMHOLE uses 17 different ortholog prediction tools in a supervised machine learning algorithm for predicting least diverged orthologs (*Sutphin et al., 2016*). The confidence scores reported are based on the number of algorithms predicting each pair, as well as the support vectors machine classifiers of the WORMHOLE algorithm—both scaled between 0 and 1 with 0.5 being the best precision-recall balance.

To our knowledge, there have not been any quantitative confidence scores of homoeolog predictions reported. However, there has been qualitative confidence reported for some polyploids. In a paper by *Cheng et al. (2012)* in the triploid *Brassica rapa* species, they gave qualitative confidence scores (high vs. low) for homoeolog pairs. Pairs were determined as high confidence if the gene was supported by transcriptome evidence, and if there was a syntenic ortholog in *Arabidopsis* (*Cheng et al., 2012*).

Here, we introduce a more fine-grained and flexible confidence score for homoeolog predictions. Based on fuzzy logic, it combines evolutionary distance, local synteny, and the extent of duplication. Fuzzy logic is about "degrees of truth" rather than binary true or false and is based on the idea that how true or not something is can be represented over a continuum. (See Table 2 for terminology related to fuzzy logic.) Most existing applications of fuzzy logic deal with control systems (*Cheng & Yeh, 1993*; *Hirulkar et al., 2014*). In our case, we can recognize homoeologs which are most certainly true predictions; we can also recognize homoeologs which are almost certainly wrong. However, the homoeolog predictions which are more dubious are harder to assign a score to.

Fuzzy logic resembles human reasoning because it works on a range of possibilities for input, expressed in linguistic terms. This is useful for practical purposes—it helps deal with uncertainty, when the lines between what is a good and bad homoeolog prediction are fuzzy.

## MATERIALS AND METHODS

In the latest OMA release, we include three agriculturally important allopolyploid crops: *Triticum aestivum* (bread wheat), *Brassica napus* (rapeseed) and *Gossypium hirsutum* (upland cotton). We used these genomes to infer homoeologous pairs of genes, with the improved confidence score assignment. We then evaluated our homoeolog predictions by comparing the confidence scores to other data aggregated from OMA, and by manually assessing a subset of the predictions.

### Polyploid genomes used

Upland cotton (*G. hirsutum*; $A_tA_tD_tD_t$; $2n = 4\times = 52$) is an important fiber crop, making up 90% of cotton production worldwide (*Zhang et al., 2015*). The diploid Gossypium progenitors diverged ~5–10 MYA, hybridized ~1–2 MYA, and was followed by a genome doubling, giving the allotetraploid cotton (*Wendel & Cronn, 2003*). The genome size of this allotetraploid is 2.25 Gb, assembled into 26 chromosomes. The "TM-1" *G. hirsutum* genome, NAU-NBI assembly and annotation v1.1 (*Zhang et al., 2015*), was obtained from ftp://ftp.bioinfo.wsu.edu/species/Gossypium_hirsutum/NAU-NBI_G.hirsutum_ AD1genome/genes/. A total of 66,967 protein-coding genes were used for homoeolog inference.

Oilseed rape (*B. napus*; $2n = 4\times = 38$) is an allotetraploid member of the Brassicaceae (mustard/cabbage) family. The progenitors of *B. napus* diverged about four MYA, followed by a relatively recent hybridization, dating back to 7,500–12,500 years ago (*Chalhoub et al., 2014*). The 1.13 Gb *B. napus* genome was sequenced and assembled into 19 pseudomolecules, with 98,130 canonical protein coding genes (*Chalhoub et al., 2014*). The "Darmor-bzh v5" version of the *B. napus* genome was obtained from (http://www.genoscope.cns.fr/brassicanapus/data/).

Wheat (*T. aestivum*) is a staple food. Its genome is large (~17 Gb) and allohexaploid, consisting of three subgenomes with seven sets of chromosomes each ($6\times = 2n = 42$). The divergence of the progenitor species is estimated to be around seven MYA, and the two hybridization events <1 MYA (*International Wheat Genome Sequencing Consortium (IWGSC), 2014*). The wheat genome was obtained from Ensembl Plants 33 using the TGACv1 assembly (*Clavijo et al., 2017*) of the "Chinese Spring" cultivar; ftp://ftp.ensembl. org/pub/release-33/embl/triticum_aestivum/. The annotation consisted of 103,458 genes after removing alternative splice variants. Canonical genes among the splice variants were chosen based on the method described in (*Altenhoff et al., 2011*).

### Homoeolog inference

In OMA, we inferred homoeologs in the polyploid species by treating each subgenome as a separate genome and inferring orthologs following the normal OMA

pipeline (*Train et al., 2017*). Briefly, the steps for obtaining pairwise homoeologs are as follows:

- **Subgenome delineation:** Polyploid genomes were separated into two or more subgenomes based on their identifiers. From here on, a subgenome is treated as a standalone genome/proteome.

- **Homology inference:** Using proteomes from all the species in OMA, Smith–Waterman alignments were made with all possible pairs of sequences from all genomes (*Smith & Waterman, 1981*). Pairs of protein sequences from different genomes (or subgenomes) with sufficient alignment score and overlap were retained.

- **Homoeolog and co-homoeolog inference:** Pairs retained from the previous step that are the mutually evolutionary closest sequences between a pair of subgenomes are kept during this step. In order to include many-to-many homoeologous relationships, pairs found within a confidence interval of the mutually closest sequences are kept.

- **Witness of "non-homoeology" verification:** In order to avoid paralogs to be mistakenly identified as homoeologs due to differential gene loss (*Dessimoz et al., 2006*), a verification step is performed by searching for a third genome that retained both orthologous copies. The third genome can be any of the 2,103 eukaryotic, prokaryotic, or archaeal genomes currently in OMA, but are often very closely related species. This third genome acts as a witnesses of non-homoeology, and pairs that do not pass this test are filtered out.

Recently, several improvements to the OMA algorithm were introduced (*Train et al., 2017*). This included refinements to account for rapidly evolving duplicated genes, as well as fragmentary sequences. Additionally, those homoeologs which were considered outliers in the previous OMA algorithm (*Altenhoff et al., 2015*) are now retained.

All data, including the genome information and homoeologous/orthologous relationships, is stored in an HDF5 database (https://omabrowser.org/All.Dec2017/OmaServer.h5), and queried programmatically using python. Our code for the confidence score computations can be found in File S1. Skfuzzy (https://github.com/scikit-fuzzy/scikit-fuzzy) was used for all functions related to fuzzy logic. Only canonical genes were considered (no alternative splice variants) for the following analyses.

## Variables used as input for fuzzy sets

Three inputs, that is, features of a given homoeolog pair, were used for the fuzzy logic variables: the evolutionary distance, synteny score, and total copy number.

Synteny is the overall conservation of chromosome order and location of genes when comparing two chromosomes. However, rearrangements may result in smaller regions of the chromosome being syntenic, rather than the whole chromosome. Thus, we computed a local synteny score for each pair of homoeologs (Fig. 1). This consisted of obtaining two windows of genes, one on each subgenome: the first one containing the 10 neighbor genes surrounding one homoeolog, and the second window containing the 10 neighbor genes surrounding the corresponding homoeolog. If the homoeolog was on a scaffold with

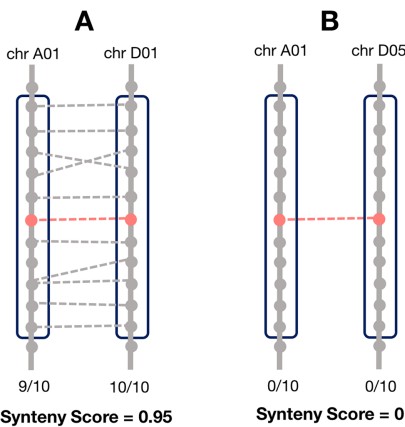

**Figure 1 Method for calculating the local synteny score for a homoeolog pair.** For each pair, a window of 10 genes surrounding each homoeolog was obtained. The synteny score is the mean proportion of genes in the windows that are homoeologous. (A) An example of a pair with a high synteny score and (B) example with a low synteny score.

less than 10 genes, all the genes on the scaffold were used. However, if there were less than two genes (including the homoeolog) on the scaffold, we assigned the synteny score to zero for these pairs. We then computed the proportion of genes (not including the homoeologs) in each window which were homoeologous to the corresponding window on the other subgenome and took the mean of these two as the synteny score.

The evolutionary distance is based on the number of nucleotide substitutions between two sequences. This is in PAM units, and calculated as part of the normal OMA algorithm (*Roth, Gonnet & Dessimoz, 2008*). A distance of one PAM unit describes an amount of evolution which will change, on average, 1% of the amino acids.

The "total copy number" is a metric to understand the degree of duplication for a pair of homoeologs. For a given pair, it is calculated as the number of homoeologs for the first gene + the number of homoeologs for the second gene.

## Defining the universe of discourse and membership functions for all variables

For each genome, input variable (synteny, evolutionary distance, total copy nr), and output variable (confidence score), the universe of discourse is the range of possible values. The universe for each variable was are defined in Table 3, using the "Antecedent" and "Consequent" functions from the skfuzzy control module. Gaussian curve membership functions were defined for each genome using the skfuzzy "gaussmf" function. The membership classes for each fuzzy set are defined in Table 4.

## Fuzzy rules and control system simulation

We created five rules based on the universes defined above and stored them in the lookup table (see Results). The control system and simulation were made using the skfuzzy control module and the rules. This simulation was then used to defuzzify, that is, return a crisp output, using the centroid defuzzify method. It takes the inputs and returns a confidence

**Table 3 Universe of discourse for all variables of the fuzzy sets.**

| Variable | Input or output | Minimum | Maximum | Step |
|---|---|---|---|---|
| Distance | Input | 0 | Distance max | 0.01 |
| Synteny score | Input | 0 | 1 | 0.01 |
| Total copy nr | Input | 2 | Total copy nr max | 1 |
| Confidence | Output | 0 | 100 | 1 |

**Table 4 Membership functions for all variables of the fuzzy sets.**

| Variable | Membership class | Central point | Standard deviation |
|---|---|---|---|
| Distance | Low | 0 | Distance maximum/10 |
| Distance | Med | Distance maximum/4 | Distance maximum/10 |
| Distance | High | Distance maximum | Distance maximum/2.5 |
| Synteny | Low | 0 | 0.15 |
| Synteny | Med | 0.3 | 0.15 |
| Synteny | High | 1 | 0.4 |
| TotalCopyNr | Low | TotalCopyNr median | TotalCopyNr median |
| TotalCopyNr | Med | $4 \times$ TotalCopyNr median | $1.5 \times$ TotalCopyNr median |
| TotalCopyNr | High | TotalCopyNr maximum | TotalCopyNr maximum/2.5 |
| Confidence | Very low | 0 | 20 |
| Confidence | Low | 50 | 10 |
| Confidence | Med | 70 | 10 |
| Confidence | High | 90 | 10 |
| Confidence | Very high | 100 | 10 |

**Note:**
Each membership class is a gaussian curve, with the central point and standard deviation defined here.

score between 0 and 100. We then kept the smallest confidence score returned as the minimum and scaled the maximum confidence score to be 100. A set of 30 homoeolog pairs were manually evaluated in *G. hirsutum*. Ten pairs were randomly chosen from the 0–60, 60–90 to 90–100 confidence score ranges. Putative functions were found by searching the protein sequence in the NCBI Conserved Domain Database (CDD) (*Marchler-Bauer et al., 2017*).

# RESULTS

## Polyploid species and homoeolog inference

We inferred homoeologs for three polyploid species using OMA: *G. hirsutum* (GOSHI), *B. napus* (BRANA), and *T. aestivum* (WHEAT). In the OMA pipeline, homoeologs are predicted as orthologs between subgenomes, thus, the genes have to be annotated as belonging to a particular subgenome in order to infer homoeology. However, as assemblies can be in different states of draft, not all genes are always mapped to a specific chromosome or subgenome. We therefore discarded 2,002, 515, and 4,151 genes from GOSHI, BRANA, and WHEAT, respectively, as they were not previously mapped to any particular subgenome. This left between 96% and 99.5% of the total genes in each
genome to infer homoeology. Using these genes, we predicted pairs of homoeologs between subgenomes for each of the three species. This resulted in 30,230, 35,661, and 88,513 pairs of homoeologs for GOSHI, BRANA, and WHEAT, respectively.

## Investigation into global versus local synteny

Rearrangements may result in smaller regions of the chromosome being syntenic, rather than the whole chromosome. In order to justify using a local synteny score rather than a global synteny based on chromosome matching between subgenomes, we computed the number of homoeologs across pairs of chromosomes between two subgenomes in each species. With OMA we predicted many homoeologs across different chromosome groups (Fig. 2).

Non-homoeologous chromosomes, that is, different chromosome groups, with increased frequency of homoeologs are consistent with known reciprocal translocations. For example, in GOSHI, there are two known large reciprocal translocations: parts of the chromosomes were exchanged between A02 and A03, as well as between A04 and A05. We would be able to see this by an increased frequency of homoeolog pairs inferred between chromosomes not belonging to the same chromosome group (Fig. 3). Indeed, with OMA we inferred 692, 1,208, 598, and 696 pairs of homoeologs between chromosomes A02/D03, A03/D02, A04/D05, and A05/D04, respectively (Fig. 2). This is significantly higher than the mean number of homoeolog pairs for non-homoeologous chromosomes. We observe even more levels of chromosomal translocation in WHEAT and BRANA as well, for example, detecting the known reciprocal translocations in WHEAT between 4A, 5A, and 7B (Ma et al., 2013).

Additionally, chromosome pairs with a few number of homoeologs may represent single-gene translocations between non-homoeologous chromosomes and should not be discarded from homoeology prediction via synteny-based methods. Taken together, these results suggest that a local synteny score is more robust than global synteny in order to account for large and small translocations.

## Description of fuzzy logic inputs

For each homoeolog pair, we used the synteny score, the evolutionary distance, and the total copy number as input.

The synteny score is the degree of local gene neighborhood conservation. Although synteny is not a hard requirement for homeologs, a conservation of synteny is a good indicator of correct homoeolog predictions. In order to account for chromosomal rearrangements, as well as genome assemblies which are not yet fully assembled into pseudomolecules, we computed a local synteny score. This technique, however, only works when both genes of a homoeolog pair have at least one neighbor gene. Therefore, we could not compute the synteny score for homoeologs that were on small scaffolds with only one gene annotated. For those pairs we set the synteny score to zero. This was 490, 0, and 36,250 pairs for GOSHI, BRANA, and WHEAT, respectively.

The evolutionary distance is based on the number of nucleotide substitutions between two sequences (in PAM units). Because of the relatively short divergence between

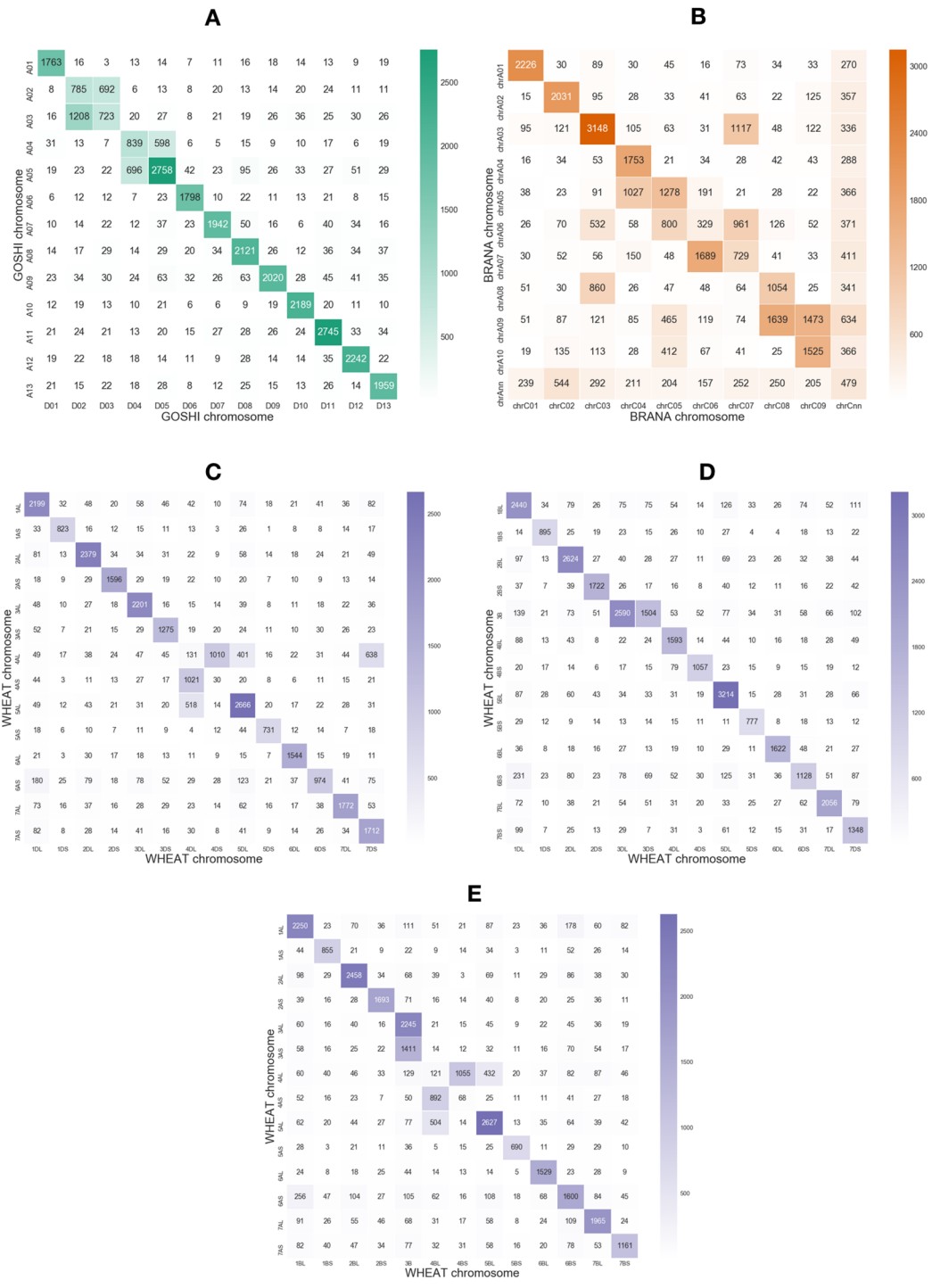

**Figure 2 Heatmaps showing the number of homoeologs predicted with OMA between each of the chromosomes in the two subgenomes in (A) GOSHI, (B) BRANA, and (C–E) WHEAT.** Homoeologs that were on scaffolds or "randoms" were mapped to their respective chromosomes. "Off-diagonal" chromosomes, that is, different chromosome groups, with an increased frequency of homoeologs are consistent with known reciprocal translocations.

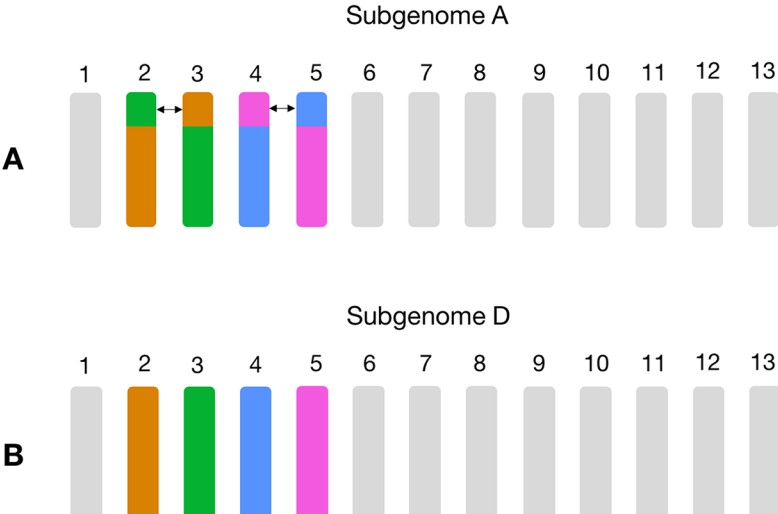

**Figure 3 Schema of the known large reciprocal translocations in GOSHI.** Chromosome segments were exchanged between A02 and A03, as well as A04 and A05 in subgenome A. This would result in an increased frequency of homoeolog pairs predicted between chromosomes A02/D03, A03/D02, A04/D05, and A05/D04, which we observe with the homoeolog pairs inferred by OMA. In this figure, chromosome segments of the same color between subgenome A (A) and subgenome D (B) are those with a high frequency of homoeolog pairs.                

subgenomes, we expect there to generally be a low distance between homoeologs. Additionally, genes which have a high number of predicted homoeologs could indicate something suspect, such as a transposable element (TE) misannoted as a gene. All distributions of the inputs are shown in Fig. 4.

## Fuzzification of data into membership functions

The membership functions allow us to translate an input value into a degree of membership between 0 and 1. The membership curves are overlapping to account for the fuzziness between categories, and for all genomes we defined what we consider to be low, medium, or high input values (Fig. 5). For example, a homoeolog pair with a synteny score of 0.6 could then be translated linguistically to not at all low synteny, somewhat medium synteny, and mostly high synteny. For synteny, the curves are statically defined based on the synteny input universe (possible scores only from 0 to 1), while the distance and total copy number are based on the minimum, maximum, and median values for that particular genome (see Materials and Methods).

## The fuzzy rules and the control output

The output from our fuzzy inference process, Confidence, also has a membership function. It is used to map the fuzzy confidence to a crisp confidence score, between 0 and 100 (Fig. 6). Fuzzy logic uses a set of IF THEN statements (rules) for mapping the input space to an output space. We set the rules as to what we think should determine the five categories of confidence (Fig. 6). The very high confidence homoeologs are those which have a high synteny, a low distance, and a low total number of homoeologs. The very low confidence homoeologs have a low synteny score, a high distance, and a high

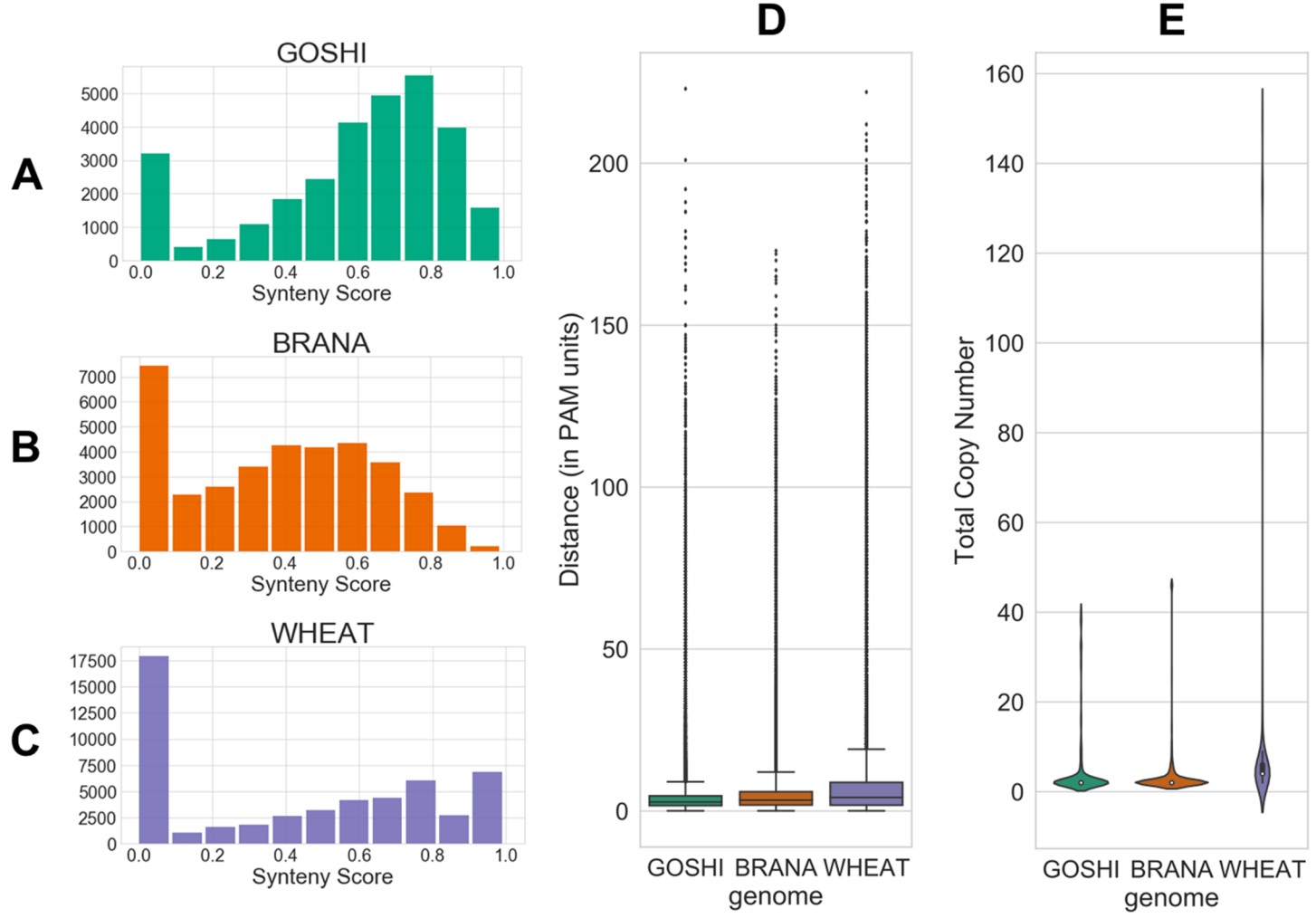

**Figure 4 Homoeolog features used for fuzzy inputs.** (A–C) Distribution of synteny scores for pairs of homoeologs for *Gossypium hirsutum*, *Brassica napus*, and *Triticum aestivum*, respectively. Pairs with one or both of the homoeologs on scaffolds with only one gene on the scaffold were removed. (D) Distributions of distance and (E) total copy number for homoeolog pairs in each genome.

copy number of homoeologs. High, medium, and low confidence is everything in between, based on what we would consider low or high inputs. We put a lot of emphasis on synteny score because the chances of a non-homoeologous pair being syntenic but not originating from a common ancestral gene is very low.

### Defuzzification

After defining the rules, we created a control system and simulation for each of the genomes. The inputs for each homoeolog pair were then fed into the simulation which contains the rules, and defuzzified. The defuzzification process converts the fuzzy linguistic confidence to a crisp confidence score, which we then scaled between the minimum value and 100. The reason for scaling to 100 is so that there would not be a sharp cutoff and a maximum confidence score around 80. This facilitates comparison of homoeolog confidence scores within genomes, as people naturally tend to associate the best score with

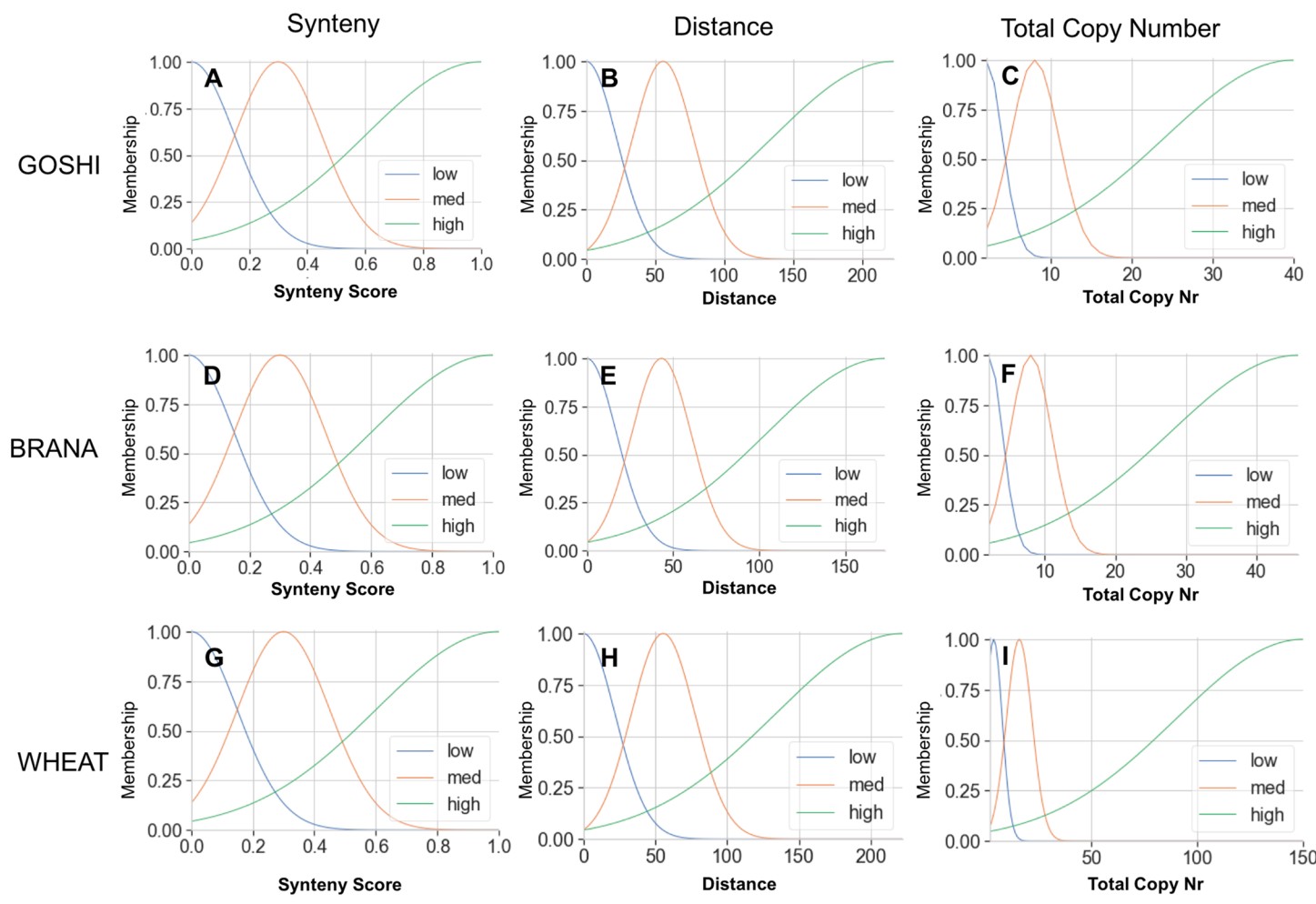

**Figure 5 Membership functions for the three inputs—synteny, distance, and total copy number—for GOSHI (A–C), BRANA (D–F), and WHEAT (G–I).**                                        

100. The resulting distribution of confidence scores is shown in Fig. 7. Unscaled confidence scores are shown in Fig. S1.

## Evaluation of homoeologs confidence scores

We assessed the homoeolog predictions by looking at the correlation between the total number of orthologs per homoeolog pair and the confidence score. We also manually evaluated a set of 10 homoeolog pairs from 0–60, 60–90, to 90–100 confidence score ranges (Table S1).

The total number of orthologs takes into account the ortholog predictions for all of the species in OMA. Homoeolog pairs with few orthologs are either lineage-specific or dubious, whereas pairs with many orthologs represent those more likely to be true. Although the correlation is low between the total number of orthologs and the confidence score ($R = 0.005$), it holds as a general trend when looking at confidence scores between 0 and 70 vs. confidence scores between 70 and 100 (Fig. 8). In the set of GOSHI

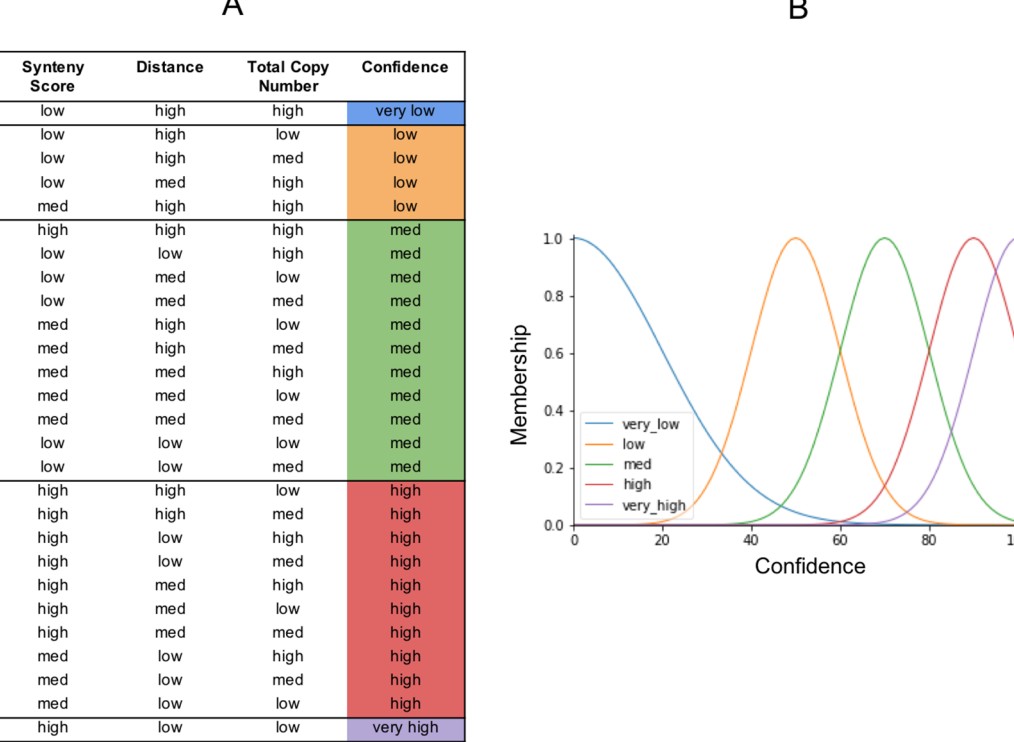

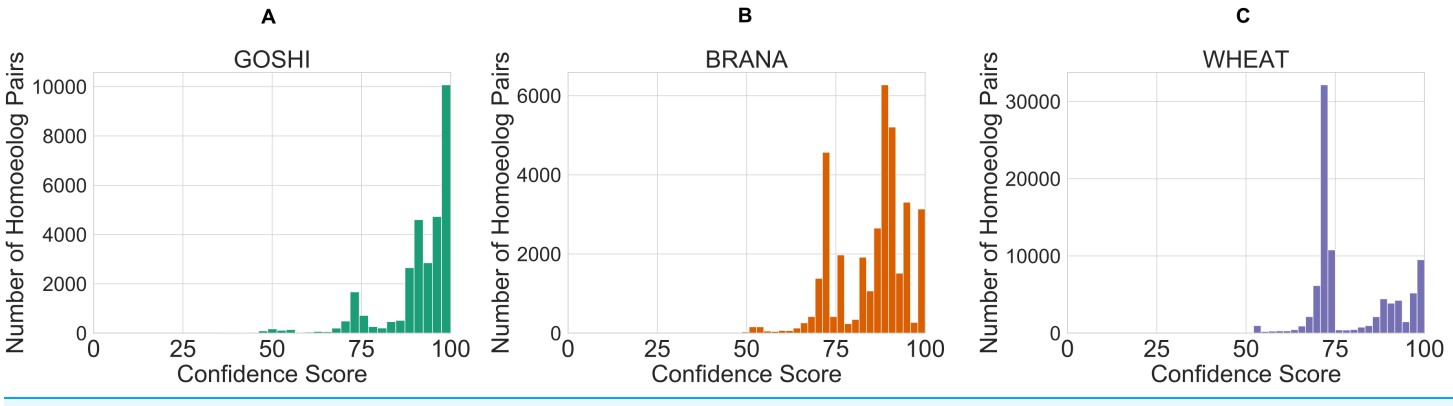

**Figure 6 The fuzzy rules.** (A) Lookup table. The first three columns are the inputs. The final column is the output (confidence), and reflects the same colors as the confidence membership curves in (B).

**Figure 7 Distribution of crisp confidence scores after scaling to 100 for GOSHI (A), BRANA (B), and WHEAT (C).**

homoeologs manually evaluated, the total number of orthologs was significantly lower for the low confidence homoeologs (0–60), than the 60–90 and 90–100 samples (Table S1).

Interestingly, in GOSHI, for the set of 10 manually evaluated pairs from 0 to 60 confidence, half had RVT-3 (reverse transcriptase-like) domains. According to the CDD description, "This domain is found in plants and appears to be part of a retrotransposon". This could explain the few number of orthologs for those with low confidence scores,
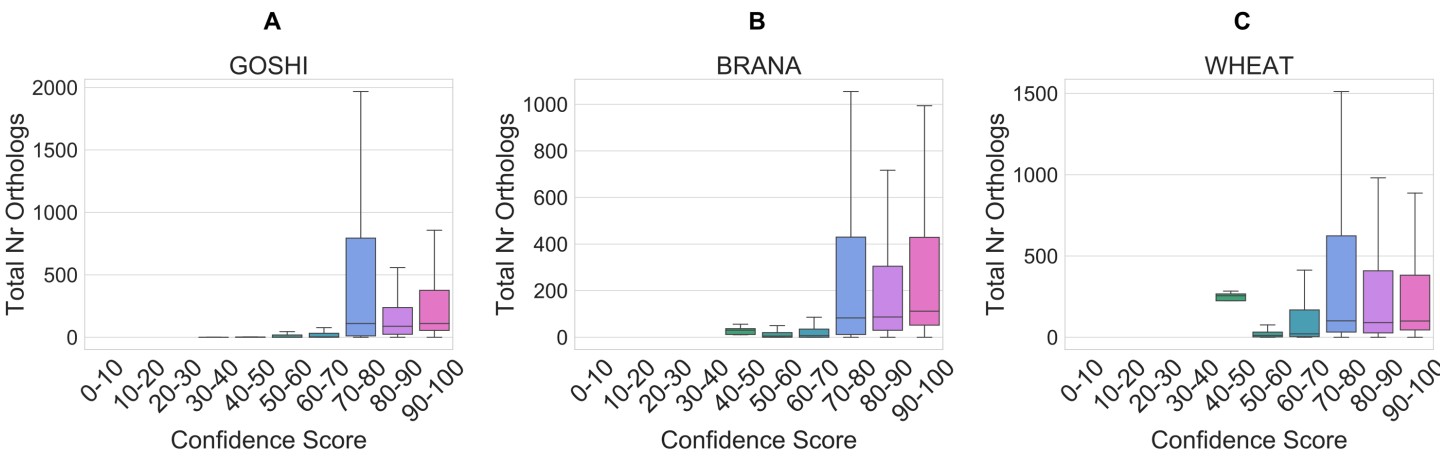

**Figure 8 Distribution of the total orthology copy number, binned by confidence score, for GOSHI (A), BRANA (B), and WHEAT (C).**

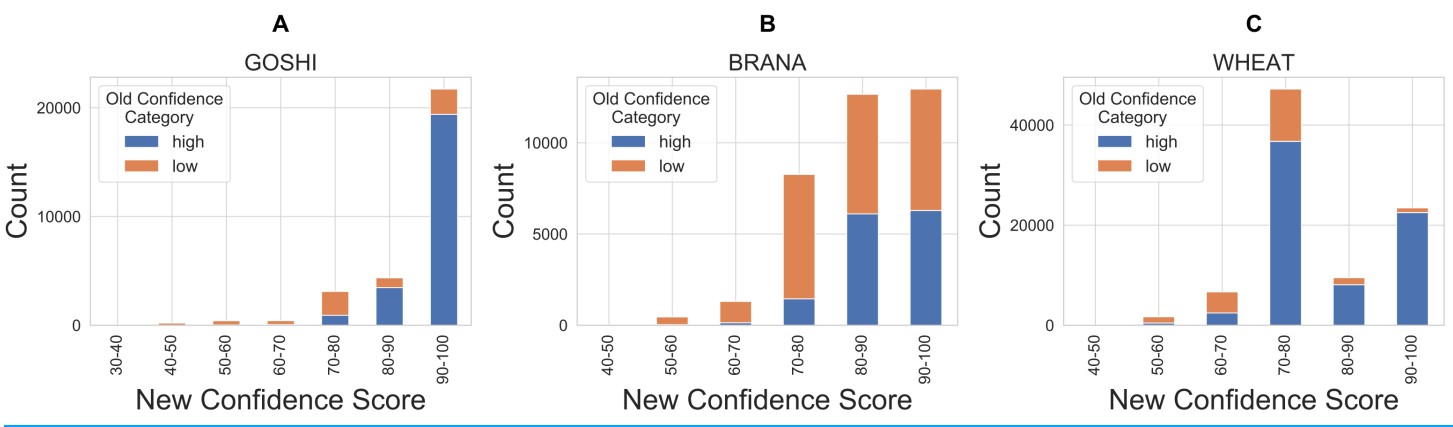

**Figure 9 Comparison of new and old confidence for GOSHI (A), BRANA (B), and WHEAT (C).**

because TEs rapidly evolve and may have lost homology in the other species. By contrast, none of the sampled homoeologs with a confidence score above 60 had RVT-3 annotation, or any functional description associated with TEs.

Finally, in order to compare our new confidence scores to the previous confidence classes in OMA, we looked at the proportion of pairs in each confidence score bin which were previously marked as either high or low confidence (Fig. 9). There are a significant amount of homoeolog pairs which were previously classified as low confidence, but are now between 90 and 100 confidence score (37%, 31%, 5% of the formerly low confidence pairs in GOSHI, BRANA, and WHEAT, respectively). On the other hand, there was a much smaller proportion of pairs that were considered high confidence before, but now have less than 70 as a confidence score (0.5%, 1%, and 4% for GOSHI, BRANA, and WHEAT, respectively). In wheat, the large peak of scores between 70 and 80 that were formerly high confidence is because 68% of these pairs are on small scaffolds for which we were unable to compute synteny; however, they belong to the same chromosome group.

## DISCUSSION

Fuzzy logic has some applications in biology (*Torres & Nieto, 2006*; *Chandgude & Pawar, 2016*), but none in the field of homology prediction or for assigning confidence scores for inferences for which a ground-truth cannot be established. Our application therefore describes an interesting and novel approach to deal with such scenarios where supervised machine learning approaches are risky to be applied.

Although our methods of defining membership functions and rules for the confidence scores may seem ad hoc, that is where fuzzy logic excels. We don't claim this method to be objective or the best schema, however, after manual evaluation, the results are interpretable and useful. An important limitation to our approach is that the confidence scores are heavily based on synteny, which is known to degrade over evolutionary time. Therefore, synteny may be low or even undetectable for older polyploids. The species used in this study are relatively young polyploids, and this approach was untested in paleopolyploids. Nevertheless, the results are relevant for the three polyploid species in OMA, and are reproducible as well, as they are coded into the OMA pipeline.

Synteny has been used already as a way to assess the confidence of ortholog pairs. For example, in Ensembl, a "gene order conservation" score uses a window of two genes on each side of a given ortholog prediction and checks whether the genes are also orthologs and in the same orientation. Furthermore, they calculate a "whole genome alignment" score which assesses the proportion of a given ortholog pair which fall within syntenic regions, with more weight given to exons that can be aligned than introns (*Aken et al., 2017*). In the SYNERGY algorithm, *Wapinski et al. (2007)* used the synteny combined with the evolutionary distance to make the ortholog predictions themselves rather than to assess ortholog predictions.

The new homoeolog confidence scores are an improvement to the old way we assigned confidence class. Going from a discrete category of high vs. low to a quantitative score can give users a wider range of options depending on the analyses they want to perform. For example, for finding differential gene expression among homoeologs, one may want to be conservative and take on those pairs with 90+ confidence. On the other hand, if scanning for all potential homoeologs that could provide disease resistance, some potential R-genes may look like highly repetitive TEs (*Bayer, Edwards & Batley, 2018*). With our fuzzy logic approach, the homoeolog pairs would have a low confidence score due to their high Total Copy Number. Thus, one could take all of the homoeologs no matter the confidence.

Between the polyploid species used in this study, there are genome specificities, biological as well as assembly-wise, which is why we see differences in terms of confidence scores. For example, wheat has the majority of its assembly still in scaffolds, which is why the peak of scores is around 70–80. These confidence scores will most likely increase when the assembly improves. However, by using the local synteny, our method at least allows us to make a confidence score using scaffolds.

Homoeologs aren't always syntenic, in one-to-one copies, or with a low distance. Pairs on non-matching chromosomes may be homoeologs that represent single-gene

translocations. Furthermore, some genes can evolve quickly, giving an abnormally high distance. Additionally, some genes might have a high copy number. These could be real genes that have a propensity for duplication (depends on the function, located in a recombination hotspot, gene balance hypothesis, etc.). It is important to not disregard these pairs in homoeolog inference methodology, as they could still represent interesting functions.

## CONCLUSION

To assign confidence scores to inferred pairs of homoeologs, we introduced a fuzzy logic-based method combining evolutionary distance, local synteny, and cardinality of homoeology relationships. Even though there is a degree of subjectivity in defining the fuzzy rules, the resulting scores proved meaningful in how they correlate with the number of orthologs and in a manual inspection of a random subset of 30 instances. The framework constitutes a substantial improvement over the previous confidence score which was only based on global synteny and had much less granularity.

### Funding

This work was supported by Swiss National Science Foundation grant 150654 and a research agreement with Bayer Crop Science NV. The funders had no role in study design, data collection and analysis, decision to publish, or preparation of the manuscript.

### Grant Disclosures

The following grant information was disclosed by the authors:
Swiss National Science Foundation: 150654.
A research agreement with Bayer Crop Science NV.

### Competing Interests

Christophe Dessimoz is an Academic Editor for PeerJ.

### Author Contributions

- Natasha M. Glover conceived and designed the experiments, performed the experiments, analyzed the data, prepared figures and/or tables, authored or reviewed drafts of the paper, approved the final draft.
- Adrian Altenhoff conceived and designed the experiments, performed the experiments, analyzed the data, contributed reagents/materials/analysis tools, authored or reviewed drafts of the paper, approved the final draft.
- Christophe Dessimoz conceived and designed the experiments, authored or reviewed drafts of the paper, approved the final draft.

### Data Availability

Data can be found in the OMA database (https://omabrowser.org/oma/archives/All.Dec2017/).

Under "Other files", select "OMA Browser database (as hdf5)".

## Supplemental Information

Supplemental information for this article can be found online at http://dx.doi.org/10.7717/peerj.6231#supplemental-information.

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
