# Peer review of "Assigning confidence scores to homoeologs using fuzzy logic"

_PeerJ, doi:10.7717/peerj.6231_

## Round 0.1 · original submission · Minor Revisions

Dear Dr. Glover and colleagues:

Thanks for submitting your manuscript to PeerJ. I apologize for the lengthy time in review, as we had trouble finding reviewers and one reviewer took a little longer than anticipated. I have now received two independent reviews of your work, and as you will see, both are rather favorable. Well done! Nonetheless, both reviewers raised some concerns about the research, and areas where the manuscript can be improved. I agree with the reviewers, and thus feel that their concerns should be adequately addressed before moving forward.

Please consider a definitions table or figure for terms that are jargon or introduced in this work. This will ensure that a broader audience can appreciate your writing.

I am recommending that you revise your manuscript accordingly, taking into account all of the issues raised by the reviewers. I do believe that your manuscript will be ready for publication once these issues are addressed.

Good luck with your revision,

-joe

Reviewer 1 ·

Basic reporting

Overall the language is clear and well-written and wholly appropriate for a scientific publication.

The background is generally well described, although the use of confidence scores in other orthology/paralogy methods could have been discussed briefly (e.g. InParanoid, but possibly others) and what approaches they use. Similarly, the authors could discuss any relevant literature showing that synteny, genetic distance or gene copy number are useful variables for assessing whether genes are homoeologs.

The layout of the material is clear and the figures are similarly clear and well labelled. (Figure 4B, should include the units for distance, i.e. PAM units)

The manuscript is coherent and self-contained.

Experimental design

The paper defines the a meaningful question (how to assign confidence scores to putative homoeologs) and the study contributes an answer to this.

The work appears to have been carried out rigorously and the method are clear.

Validity of the findings

I have some doubts about the language used to describe the findings. These are relatively minor, affecting one statement in the abstract and a couple of expansions on this statement in the main text. It doesn't have any impact on the validity of the paper but I'd suggest removing the claims, as detailed below:

The method is developed and the results (the confidence score) compared to an 'independent metric' (line 258), the total number of orthologs for the putative homoeolog pair. The authors reason: "Homoeolog pairs with few orthologs are either lineage-specific or dubious, whereas pairs with many orthologs represent those likely to be true". However, they find the correlation between their confidence score and this measure to be R=0.006. In the abstract the authors say the metric corroborates their confidence scores. In my view, such a low correlation shows that either the confidence scores are a very poor measure or the independent metric is. I don't believe it corroborates them.

The authors also do a manual evaluation (Supplemental Table 1 and lines 264-270). I thought this was a lot better for assessing the described confidence score and was useful to the reader. The comparison with the old scores from 271-281 is also useful.

My suggestion is that the reason for the very poor correlation is that the independent metric is actually poor, not their confidence measure. I think the independent measure can be included in the paper but I don't think the claims made from it are valid. Figure 8 is used to suggest that despite the almost zero correlation that their is a general trend. There aren't any statistical tests to confirm this. By eye, the figure seems to suggest two classes, 'low' and 'high', like their old measure. As such, I think the authors should remove the claims based on this independent measure. I'd suggest - not saying in the abstract that the independent metric corroborates the confidence score - not calling it a 'benchmark' on line 77 (it isn't used as a benchmark) - not saying on line 317 "the resulting scores proved meaningful in how they correlate with the number of orthologs"

Additional comments

Overall I thought the paper was clearly written and addressed the research question posed. The methodolgy was sound. There are a couple of minor points that could be addressed:

Figure 5: The synteny membership function for 'high' decreases as the synteny scores increases above ~0.7 for all species. This means that a score of 1.0 belongs to the 'high synteny' class less strongly than a synteny score of 0.8. I don't believe this is correct. I think the membership function should be redefined to avoid this or the authors should justify in the text why it should be this way. This problem is avoided for the 'distance' and 'total copy-number' variables.

Line 162 Should also reference figure 5 I believe

167-168. The text here is unclear/confusing. "Confidence scores were then scaled between the minimum score and 100.". This needs to be stated more clearly. It sounds like 0 was mapped to 'the minimum score' (but then what the 'minimum score' is would be undefined). My best guess is that the opposite was done: the crisp scores were linearly mapped with the minimum crisp score mapped to zero and 100 mapped to 100. Can this part of the method be stated more clearly please.

167-168. A more serious question related to this? Why were the confidence scores produced by the fuzzy logic rules subsequently re-scaled at all? This seems to go against the point of defining the fuzzy logic system. The rules, as written, already provided the confidence scores in the range 0-100. As such what OMA reports is no longer the fuzzy logic confidence scores from their system but a re-scaled version of them. It seems like the authors were not happy with the confidence scores returned and so re-scaled them? This should be explained.

Figure 7 are these the fuzzy logic confidence scores or the re-scaled ones?

220. It seems like we don't know that the errors in assignment will be only at the high-end of the range of sequence distances, not also at the low range (e.g. from more recent duplications than the divergence in question)? The text says "we expect there to generally be a low distance between homoeologs", but this is achieved by scaling the membership function to the general range of observed divergence times (Table 2). So the highest confidence for divergence at the initial speciation would be for those in the middle of the range and lower confidence would apply to both those that are older and younger than expected, I would have thought? Perhaps the authors could address this?

227: "would could"

Reviewer 2 ·

Basic reporting

Glover, Altenhoff, and Dessimoz present a very interesting approach to the issue of homoeolog identification. The use of fuzzy logic to assign confidence scores sheds light on the fact that the identity of these genes is not always known. This is an important point that would benefit the polyploid community and beyond.

The article has some issues with accessibility in the writing. As a evolutionary biologist who studies polyploidy and does quite a bit of bioinformatics, I found aspects of the fuzzy logic description difficult to follow. Some of this could be easily alleviated by explanation of terms or acronyms (like PAM units, line 150). Another example would be line 158. Clarification of "universe" here would be helpful. Is it the range of input variables? Output variables? Both? The use of "fuzzification" and "defuzzification" throws me off a bit, though I am sure they are perfectly reasonable for use when discussing fuzzy logic. Here, again, a quick definition would be very helpful.

Experimental design

The design is fine but the limitation that these are all younger polyploidy (mesopolyploids) would be useful to be addressed. Since there is a syntenic factor to the confidence score, older events might have fewer identified homoeologs. This potentially limits the scope of use for this approach, though this exact limitation is not test nor mentioned. It would be useful for the authors (probably in discussion) to speculate on these limits.

Validity of the findings

The data/results seem to be very clear with what the authors claim. I especially like that they say that this is not necessarily the best method or schema, but that, based on their manual curation, it is an improvement.

The approach is limited to genomes where the subgenomes are assigned. This is not always going to be possible. The authors acknowledge this.

---

## Round 0.2 · accepted · Accept

Dear Dr. Glover and colleagues:

Thanks for re-submitting your manuscript to PeerJ, and for addressing the concerns raised by the reviewers. I now believe that your manuscript is suitable for publication. Congratulations! I look forward to seeing this work in print, and I anticipate it being an important resource for genomics researchers. Thanks again for choosing PeerJ to publish such important work.

-joe

#

---

## Author Rebuttal · Round 0.2

Dear Joseph,

First off apologies for the delay in resubmitting the manuscript. I believe now we have fully integrated yours and the reviewers suggestions, which required a near complete redo of the analyses (after changing the definition of our synteny membership function). I think it's good to go now! We have responded in the remaining part of this letter to each reviewer comment, highlighted in red.

I would also like to thank you for the suggestion of putting a table of definitions. We have now two tables to define jargon-- one table for the fuzzy logic terms, and another table for biology terms (mostly related to homoeology). This is in hopes to make the paper clear to non-specialists coming from both fields. Theses tables could be included in a glossary, for now we have put them as Tables 1 & 2.

Thanks again for your consideration of this manuscript.

Best,
Natasha

| Reviewer 1 (Anonymous) |
| --- |
| **Basic reporting** |
| Overall the language is clear and well-written and wholly appropriate for a scientific publication.<br><br>The background is generally well described, although the use of confidence scores in other orthology/paralogy methods could have been discussed briefly (e.g. InParanoid, but possibly others) and what approaches they use. Similarly, the authors could discuss any relevant literature showing that synteny, genetic distance or gene copy number are useful variables for assessing whether genes are homoeologs.<br>This is indeed an oversight, so we added more citations in the Introduction about quantitative confidence scores in the literature. We also added information from the literature to the Discussion about using synteny to asses ortholog predictions. However, we were not able to find any papers discussing using evolutionary distance or copy number for assessing ortholog or homoeolog predictions.<br><br>The layout of the material is clear and the figures are similarly clear and well labelled. (Figure 4B, should include the units for distance, i.e. PAM units)<br>We added PAM units to axis label.<br><br>The manuscript is coherent and self-contained. |

| Experimental design |
| --- |
| The paper defines the a meaningful question (how to assign confidence scores to putative homoeologs) and the study contributes an answer to this.<br><br>The work appears to have been carried out rigorously and the method are clear. |

| Validity of the findings |
| --- |
| I have some doubts about the language used to describe the findings. These are relatively minor, affecting one statement in the abstract and a couple of expansions on this statement in the main text. It doesn't have any impact on the validity of the paper but I'd suggest removing the claims, as detailed below:<br><br>The method is developed and the results (the confidence score) compared to an 'independent metric' (line 258), the total number of orthologs for the putative homoeolog pair. The authors reason: "Homoeolog pairs with few orthologs are either lineage-specific or dubious, whereas pairs with many orthologs represent those likely to be true". However, they find the correlation between their confidence score and this measure to be R=0.006. In the abstract the authors say the metric corroborates their confidence scores. In my view, such a low correlation shows that either the confidence scores are a very poor measure or the independent metric is. I don't believe it corroborates them.<br><br>We agree that the correlation is quite low, so we removed this line from the abstract.<br><br>The authors also do a manual evaluation (Supplemental Table 1 and lines 264-270). I thought this was a lot better for assessing the described confidence score and was useful to the reader. The comparison with the old scores from 271-281 is also useful.<br><br>We thank the reviewer for the supportive statements.<br><br>My suggestion is that the reason for the very poor correlation is that the independent metric is actually poor, not their confidence measure. I think the independent measure can be included in the paper but I don't think the claims made from it are valid. Figure 8 is used to suggest that despite the almost zero correlation that their is a general trend. There aren't any statistical tests to confirm this. By eye, the figure seems to suggest two classes, 'low' and 'high', like their old measure. As such, I think the authors should remove the claims based on this independent measure. I'd suggest - not saying in the abstract that the independent metric corroborates the confidence score - not calling it a 'benchmark' on line 77 (it isn't used as a benchmark) - not saying on line 317 "the resulting scores proved meaningful in how they correlate with the number of orthologs" |

Thanks for voicing your concern about how we describe using the number orthologs for a given homoeolog pair as a way to assess the quality. Since the correlation is quite low, we removed the line in the abstract saying it is an independent metric to corroborate the results, and we also toned down the language in the text so as to not call it a benchmark. We kept the result in the paper however, as it does show that low confidence homoeologs tend to have few orthologs as well. We changed the text to say: "Although the correlation is low between the total number of orthologs and the confidence score (R=0.005), **it holds as a general trend when looking at confidence scores between 0-70 versus confidence scores between 70-100** (Figure 8)."

Comments for the Author

Overall I thought the paper was clearly written and addressed the research question posed. The methodolgy was sound. There are a couple of minor points that could be addressed:

Figure 5: The synteny membership function for 'high' decreases as the synteny scores increases above ~0.7 for all species. This means that a score of 1.0 belongs to the 'high synteny' class less strongly than a synteny score of 0.8. I don't believe this is correct. I think the membership function should be redefined to avoid this or the authors should justify in the text why it should be this way. This problem is avoided for the 'distance' and 'total copy-number' variables.

Thank you for pointing out this oversight. This is changed now so that the synteny membership function is at its peak at 1.0. The results have been recalculated and are slightly different now, but similar to before.

Line 162 Should also reference figure 5 I believe

This is true, but we didn't add figure 5 at this because it would make the reference to figure 5 before figure 2. We decided to keep figure 5 in the results rather than the materials and methods so we think just referencing table 2 is sufficient.

167-168. The text here is unclear/confusing. "Confidence scores were then scaled between the minimum score and 100.". This needs to be stated more clearly. It sounds like 0 was mapped to 'the minimum score' (but then what the 'minimum score' is would be undefined). My best guess is that the opposite was done: the crisp scores were linearly mapped with the minimum crisp score mapped to zero and 100 mapped to 100. Can this part of the method be stated more clearly please.

We now try to be more clear in describing how and why we scaled the confidence scores. We changed text to: "It takes the inputs and returns a confidence score between 0-100. We then kept the smallest confidence score returned as the minimum and scaled the maximum confidence score to be 100."

167-168. A more serious question related to this? Why were the confidence scores produced by the fuzzy logic rules subsequently re-scaled at all? This seems to go against the point of defining the fuzzy logic system. The rules, as written, already provided the confidence scores in the range 0-100. As such what OMA reports is no longer the fuzzy logic confidence scores from their system but a re-scaled version of them. It seems like the authors were not happy with the confidence scores returned and so re-scaled them? This should be explained.

This is a serious point and we appreciate the reviewer's concern. In a way it is true that we were not happy with the unscaled confidence scores. This is because no matter how we slightly changed the membership functions, the maximum crisp confidence score was around 80-90. However, when considering from the end-user, intuitive perspective, humans automatically equate "100" to the "best" score. We added in the text: "The reason for scaling to 100 is so that there would not be a sharp cutoff and a maximum confidence score around 80. This facilitates comparison of homoeolog confidence

scores within genomes, as people naturally tend to associate the best score with 100. The resulting distribution of confidence scores is shown in Figure 7. Unscaled confidence scores are shown in Supplementary Figure 1."

Figure 7 are these the fuzzy logic confidence scores or the re-scaled ones?
These are rescaled confidence scores. We now reflect this in the figure legend and also added a supplementary figure which shows the unscaled confidence scores.

220. It seems like we don't know that the errors in assignment will be only at the high-end of the range of sequence distances, not also at the low range (e.g. from more recent duplications than the divergence in question)? The text says "we expect there to generally be a low distance between homoeologs", but this is achieved by scaling the membership function to the general range of observed divergence times (Table 2). So the highest confidence for divergence at the initial speciation would be for those in the middle of the range and lower confidence would apply to both those that are older and younger than expected, I would have thought? Perhaps the authors could address this?

For the distance membership, the "low" distance category always peaks around 0 (Figure 5). Most homoeologs have a distance close to 0 (median 3.5). We do not take into account those pairs which may have duplicated after the speciation event (paralogs mistaken as homoeologs) because we think most of them have been removed as the normal "witness of nonorthology" step as part of the OMA algorithm.

227: "would could"
Changed.

Reviewer 2 (Anonymous)

Basic reporting

Glover, Altenhoff, and Dessimoz present a very interesting approach to the issue of homoeolog identification. The use of fuzzy logic to assign confidence scores sheds light on the fact that the identity of these genes is not always known. This is an important point that would benefit the polyploid community and beyond.

The article has some issues with accessibility in the writing. As a evolutionary biologist who studies polyploidy and does quite a bit of bioinformatics, I found aspects of the fuzzy logic description difficult to follow. Some of this could be easily alleviated by explanation of terms or acronyms (like PAM units, line 150). Another example would be line 158. Clarification of "universe" here would be helpful. Is it the range of input variables? Output variables? Both? The use of "fuzzification" and "defuzzification" throws me off a bit, though I am sure they are perfectly reasonable for use when discussing fuzzy logic. Here, again, a quick definition would be very helpful.

Thank you for your comment. We have added in a definitions table (Supplementary tables) to help make the jardon clear to both biologists and mathematicians alike.

## Experimental design

The design is fine but the limitation that these are all younger polyploidy (mesopolyploids) would be useful to be addressed. Since there is a syntenic factor to the confidence score, older events might have fewer identified homoeologs. This potentially limits the scope of use for this approach, though this exact limitation is not test nor mentioned. It would be useful for the authors (probably in discussion) to speculate on these limits.

This is indeed true. Therefore we added in the Discussion: "An important limitation to our approach is that the confidence scores are heavily based on synteny, which is known to degrade over evolutionary time. Therefore, synteny may be low or even undetectable for older polyploids. The species used in this study are relatively young polyploids, and this approach was untested in paleopolyploids. Nevertheless, the results are reproducible, as they are coded into the OMA pipeline."

## Validity of the findings

The data/results seem to be very clear with what the authors claim. I especially like that they say that this is not necessarily the best method or schema, but that, based on their manual curation, it is an improvement.

The approach is limited to genomes where the subgenomes are assigned. This is not always going to be possible. The authors acknowledge this.
We thank the reviewers for taking the time to review the manuscript and provide their input.